Review article

# The genetic basis of autoimmunity seen through the lens of T cell functional traits

Kaitlyn A. Lagattuta[1,2,3,4,5], Hannah L. Park [1,2,3,4,5], Laurie Rumker[1,2,3,4,5], Kazuyoshi Ishigaki [1,2,3,4,6], Aparna Nathan [1,2,3,4,5,7] ✉ & Soumya Raychaudhuri [1,2,3,4,5,7] ✉

Autoimmune disease heritability is enriched in T cell-specific regulatory regions of the genome. Modern-day T cell datasets now enable association studies between single nucleotide polymorphisms (SNPs) and a myriad of molecular phenotypes, including chromatin accessibility, gene expression, transcriptional programs, T cell antigen receptor (TCR) amino acid usage, and cell state abundances. Such studies have identified hundreds of quantitative trait loci (QTLs) in T cells that colocalize with genetic risk for autoimmune disease. The key challenge facing immunologists today lies in synthesizing these results toward a unified understanding of the autoimmune T cell: which genes, cell states, and antigens drive tissue destruction?

Genetic risk variants for autoimmune disease are enriched in T cell-specific regulatory regions of the genome[1,2] and are disproportionately close to genes with T cell-specific functions[3]. These findings imply that T cells are critically important to the development of autoimmune disease. To understand how T cells change to enact autoimmune destruction, we can identify quantitative trait loci (QTLs). A QTL is a location in the genome in which someone's DNA sequence helps to explain some quantifiable characteristic of the individual. These characteristics range from general traits such as birthweight[4] to detailed molecular traits such as the expression level of a certain gene. By using molecular sequencing data, we can now conduct QTL studies to search for the T cell molecular traits (e.g., gene expression, chromatin accessibility) that mediate risk for autoimmunity.

An expression QTL (eQTL) is a QTL that explains a statistically significant proportion of variance in the expression of a gene. The mechanism for this change in gene expression largely depends on how close the eQTL is to the target gene. Located near their target genes, cis-eQTLs impact expression often by altering transcription factor binding in a proximal regulatory element[5], or altering the rate of mRNA degradation[6]. By contrast, trans-eQTLs may regulate distant gene targets by first altering the expression of a nearby transcription factor (cis-eQTL mediation)[7]. However, trans-eQTLs can also emerge from unusual mechanisms. For example, trans-eQTLs in the major histocompatibility (MHC) locus on chromosome 6 ultimately affect the expression of TCR genes on chromosomes 7 and 14[8]. Most likely, these trans-eQTLs alter which antigenic peptides can be bound and presented by HLA, which in turn shapes the thymic selection of TCRs[9].

eQTLs may help to identify genes that mediate genetic risk for autoimmune disease. For example, rs3087243 is a single-nucleotide polymorphism (SNP) (reference allele: "G", alternate allele: "A") near the gene for CTLA4, a vital negative regulator of T cell activation[10]. An individual can have 0, 1, or 2 copies of the "G" allele. Since each additional copy of the "G" allele corresponds to a decrease in the expression of the nearby gene CTLA4[11,12], rs3087243 is a cis-eQTL. The "G" allele for rs3087243 not only corresponds to decreased expression of CTLA4, but also elevated risk for Graves' disease, rheumatoid arthritis (RA), and type-1 diabetes (T1D)[13–15]. This overlap at rs3087243 suggests that decreased expression of CTLA4 is in part responsible for autoimmune destruction in Graves', RA, and T1D. Analogously, trans-eQTLs for TCR genes raise the possibility that the MHC locus shapes risk for autoimmunity by promoting the thymic selection of autoreactive TCRs[9]. In this review, we will examine recent approaches to identify T

[1]Center for Data Sciences, Brigham and Women's Hospital, Harvard Medical School, Boston, MA, USA. [2]Division of Genetics, Department of Medicine, Brigham and Women's Hospital and Harvard Medical School, Boston, MA, USA. [3]Division of Rheumatology, Inflammation, and Immunity, Department of Medicine, Brigham and Women's Hospital and Harvard Medical School, Boston, MA, USA. [4]Program in Medical and Population Genetics, Broad Institute of MIT and Harvard, Cambridge, MA, USA. [5]Department of Biomedical Informatics, Harvard Medical School, Boston, MA, USA. [6]Laboratory for Human Immunogenetics, RIKEN Center for Integrative Medical Sciences, Yokohama, Japan. [7]These authors contributed equally: Aparna Nathan, Soumya Raychaudhuri. ✉e-mail: aparna_nathan@hms.harvard.edu; soumya@broadinstitute.org

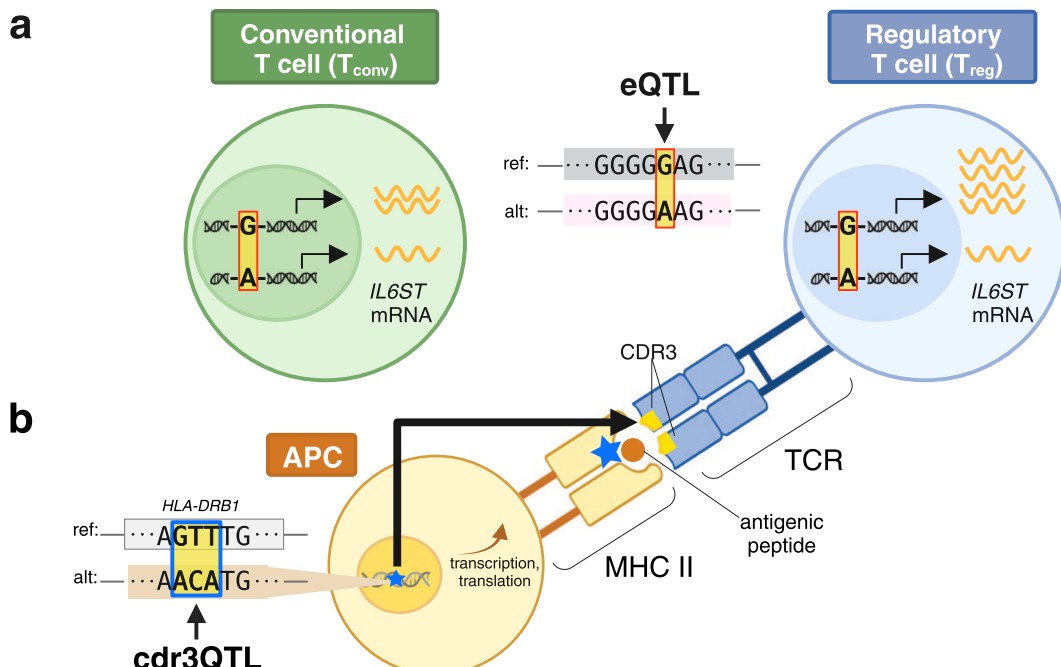

**Fig. 1 | A schematic of quantitative trait loci (QTLs) affecting T cell functional traits. a** A T cell state-dependent expression QTL (eQTL) at rs7731626 is depicted in two different T cells: one in a conventional T cell state (T_conv, left) and one in a regulatory T cell state (T_reg, right). Based on eQTL studies, we expect the "G" risk allele for rheumatoid arthritis and multiple sclerosis to increase *IL6ST* expression to a greater extent in the T_reg compared to the T_conv. **b** The associated molecular trait for a cdr3QTL is amino acid usage in the T cell receptor (TCR). In this antigen presenting cell (APC), *HLA-DRB1* is transcribed, translated, and loaded with antigen while harboring a genetic variant at amino acid position 13. This genetic variant in *HLA-DRB1* appears to affect amino acid usage in the TCR, by influencing which T cells survive in the thymus. The blue star indicates the location of the cdr3QTL, at the level of *HLA-DRB1* DNA as well as HLA-DRB1 protein. QTL quantitative trait locus, ref reference, alt alternative, CDR3 complementarity determining region 3, MHC major histocompatibility complex. Created with BioRender.com.

cell QTLs, and discuss how to connect numerous types of T cell QTLs to autoimmune disease.

## Expression quantitative trait loci (eQTLs): the importance of T cell state

Unfortunately, most risk variants for autoimmune disease do not coincide with the eQTLs identified through bulk RNA sequencing (RNAseq)[16]. While rs3087243 clearly implicates *CTLA4* gene expression in RA pathogenesis, cases such as these are quite infrequent. Accounting for linkage disequilibrium (LD), only 25% of neighboring (<100 kb) autoimmune and eQTL associations actually appear to share underlying genetic causes ("statistical colocalization"), using bulk RNAseq data[16].

It is possible that eQTLs relevant to disease only occur in some cells of a given sample. If so, eQTL effects within these pathogenic cells may become diluted and undetectable when all cell types are mixed together for bulk RNAseq. Consistent with this hypothesis, dividing bulk RNAseq tissue samples into constituent cell types by flow cytometry[17] or in silico deconvolution[18] has nominated more eQTLs that are significantly enriched for disease loci.

Single-cell RNAseq (scRNAseq) data allows researchers to accurately categorize and analyze the different cell types present in a sample. With scRNAseq, recent studies have confirmed that eQTL effects can depend on the type of cell under study. For example, in a study of memory T cells sampled from the peripheral blood of 259 individuals, we observed that additional copies of the "G" allele for the RA-associated SNP rs7731626 corresponded to a larger increase in *IL6ST* expression in regulatory T (T_reg) cells compared to non-regulatory T cells[12] (Fig. 1a). Evidence that the functional consequences of rs7731626 may be concentrated in T_reg cells could extend to many other autoimmune-associated SNPs, which are known to be enriched in certain naïve T_reg-specific regulatory regions of the genome[1,2,19]. T_reg cells, while highly relevant to immune-mediated disease, constitute a rare cell state (<5% of peripheral T cells[20]). Thus, it will be important to continue focused isolation and analysis of the T_reg cell population for eQTL discovery[21].

There are multiple ways to represent how eQTL effects depend on transcriptional context. Our research group uses dimensionality reduction to identify groups of genes whose expression changes together, forming a gradient across cells. These transcriptional gradients often approximate the extent of a known T cell state, such as cytotoxicity. Our research group then identifies cell-state-dependent eQTLs, where a genotype's effect on the expression of a gene varies along the transcriptional gradient[12]. However, because transcriptional gradients may correlate with key marker genes, it is sometimes possible to reframe cell-state-dependent eQTLs as co-expression QTLs (co-eQTLs)[22,23]. In the co-eQTL framing, the correlation in expression between gene A and gene B depends on a genotype. An alternative way to describe this phenomenon is that the genotype's effect on gene A depends on the expression of gene B. If gene B is expressed in a specific cell state, cell-state-dependent eQTL and co-eQTL are synonymous terms. If gene B does not tag a cell state, the locus is not a cell-state-dependent eQTL, but would still be considered a co-eQTL. The co-eQTL framework, therefore, offers a broader definition of gene expression interaction. The vast number of possible pairings between genes and their regulatory loci precludes comprehensive detection of co-eQTLs, which would be required to estimate the proportion of eQTLs that are co-eQTLs. Alternatively, by focusing on cell states rather than individual genes, we were able to estimate that approximately one third (33%) of eQTLs in T cells depend on transcriptional state[12].

Recent studies have estimated that a substantial proportion of eQTLs depends on cell state, and that this proportion increases when considering autoimmune disease loci. Soskic*, Cano-Gamez* et al.[24]

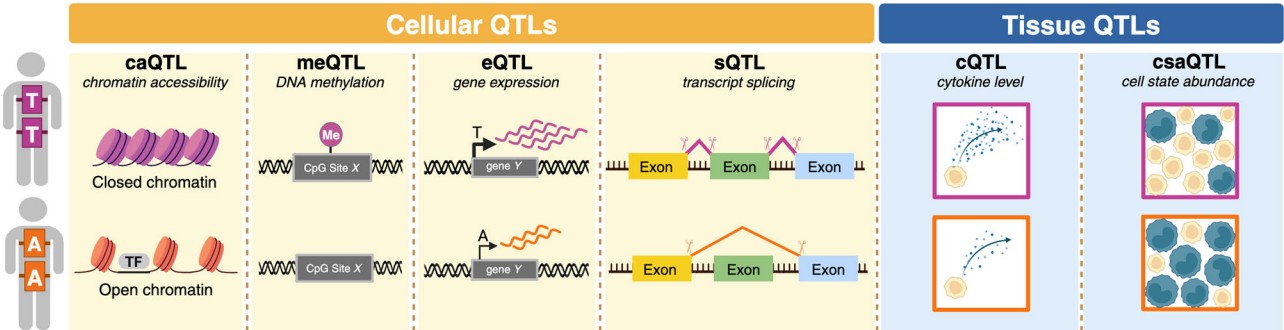

**Fig. 2 | Quantitative Trait Loci (QTLs) can regulate diverse molecular phenotypes.** Schematic illustrating molecular phenotypes that could be affected by a hypothetical quantitative trait locus (QTL). At this hypothetical locus, an individual may have a "TT" genotype (in purple), an "AA" genotype (in orange), or be heterozygous. Along the top row, we see molecular consequences of genotype "TT". Along the bottom row, we see molecular consequences of genotype "AA." We depict six types of QTLs as examples; this set of six is not comprehensive. From left to right: For the hypothetical caQTL, individuals with the "TT" genotype exhibit more closed chromatin than individuals with the "AA" genotype. For the hypothetical meQTL, CpG site X tends to be methylated in individuals with the "TT" genotype, but tends to be unmethylated in individuals with the "AA" genotype. For the hypothetical eQTL, individuals with the "TT" genotype exhibit greater expression of gene Y compared to individuals with the "AA" genotype. For the hypothetical sQTL, RNA splicing in individuals with the "TT" genotype retains all exons in the transcript of gene Y, while RNA splicing in individuals with the "AA" genotype excludes the middle exon from the transcript of gene Y. For the hypothetical cQTL, individuals with the "TT" genotype exhibit higher levels of cytokine Z (in blue) compared to individuals with the "AA" genotype. For the hypothetical csaQTL, individuals with the "TT" genotype exhibit a greater relative abundance of the yellow cell state compared to individuals with the "AA" genotype. Me methylation, TF transcription factor. Created with Biorender.com.

profiled CD4[+] T cells over a time course of anti-CD3/anti-CD28 stimulation, and found that 2265 of 6407 (35%) eQTLs depended on the activation state of the T cell. They then queried whether each eQTL statistically colocalized with genetic risk for immune-mediated disease. 60% of the colocalizing eQTLs were specific to T cell activation, and would have been missed if they did not account for this cell state dependence[24]. In a similar study, Yazar*, Alquicira-Hernandez*, Wing* et al.[25] applied scRNAseq to PBMCs at steady state, and scanned for eQTLs that colocalized with genetic risk for autoimmune disease. They observed that 68% of the T cell eQTLs colocalizing with disease loci were detected in only one of the T cell states[25]. Evidently, accounting for T cell states has substantially enhanced disease-relevant eQTL discovery in T cells. The majority of autoimmune GWAS associations, however, still remain unexplained.

## Genetic regulation of T cell functions

While gene expression is an important molecular trait, any quantity that varies across people and is measured by molecular assays can be studied as a quantitative molecular trait. Expansion of multimodal single cell technologies has prompted an ever-growing list of quantitative molecular traits and genomic loci that regulate them. Other types of QTLs are steadily gaining recognition, such as chromatin accessibility QTLs (caQTLs), histone modification QTLs (hQTLs), DNA methylation QTLs (meQTLs), splicing QTLs (sQTLs), and protein QTLs (pQTLs) (Fig. 2). These QTLs are conceptually analogous to eQTLs, with gene expression substituted for some other molecular trait; particular approaches are comprehensively reviewed elsewhere[26]. We and others have identified cell morphology QTLs (cmQTLs)[27] by measuring thousands of morphological phenotypes (e.g. size of mitochondria, granularity of endoplasmic reticulum) in vitro through multiplexed staining and imaging. cmQTLs can connect genetic variants to cellular function in human-derived, disease-relevant cell types.

In general, QTL studies query thousands of molecular traits simultaneously. Such investigations are relatively unbiased, data-driven, and can be readily applied to any cell type. However, the number of hypotheses considered imposes a substantial statistical multiple testing burden. As a result, studies often limit eQTL searches to SNPs within 1 Mb of the target gene (cis-eQTLs), and are only powered to detect eQTLs with relatively large effects[28].

For T cells, can we search for QTLs in a way that takes advantage of known T cell functions? In the 1970's, the observation that activated

lymphocytes secrete proteins which markedly impact other lymphocytes spurred a series of supernatant studies, identifying soluble cytokines[29]. These formative studies led to a common framework, in which the immune system can be understood through cytokine signaling. Cytokine signaling facilitates important T cell functions, including the attack of foreign antigens—or self-antigens, in the case of autoimmunity. Thus, identifying cytokine QTLs (cQTLs) may be an efficient and biologically interpretable way to search for disease-relevant molecular traits.

In a recent demonstration of the power of cQTLs, Nath et al.[30] analyzed 11 circulating cytokines, each of which may functionally represent the coordinated activity of thousands of genes. By starting with a curated set of molecular traits ($n = 11$), Nath et al. had enough statistical power to scan the entire genome and find genetic regulation in *trans*. Rather than applying a univariate GWAS framework to each of the 11 cytokines separately, Nath et al. considered the 11 cytokines jointly through a multivariate GWAS framework. The multivariate framework finds an optimal linear combination of input traits (cytokine levels) regulated by each locus, a powerful strategy to detect QTLs from potentially redundant trait measurements. This approach identified eight cQTLs, one of which significantly colocalizes with genetic risk for ulcerative colitis (UC)[30].

An alternative way to leverage immunological knowledge in the search for disease-relevant QTLs is to focus on cell state abundance QTLs (csaQTLs). A recent large-scale csaQTL study[31] analyzed 118 immune cell populations isolated by canonical surface markers in 3357 individuals. Orrù et al. queried csaQTLs as well as pQTLs and cmQTLs, resulting in 731 molecular traits. Of the 122 significant association signals, 51 colocalized with genetic risk for at least one autoimmune disease. For example, rs72928038 near *BACH2* was associated with increased CD28 expression on CD45RA[+] cells (predominantly naïve T cells) and was in high LD with risk alleles for autoimmune thyroiditis, vitiligo, multiple sclerosis (MS) and T1D. Focusing on T cell subpopulations rather than gene expression helps to reduce the number of molecular traits under investigation, thereby achieving enough statistical power to detect some QTLs in *trans*. In fact, the majority of significant associations reported by Orrù et al. corresponded to SNPs and cell state protein markers from different chromosomes[31]. However, this sort of study design is restricted to pre-specified T cell states and functions, since protein markers and gates need to be defined a priori.

It is now possible to define cell states in the context of high-dimensional single-cell data. For example, RNA sequencing has enabled data-driven strategies to refine our understanding of T cell states. We have found canonical correlation analysis between mRNA and protein to be effective in identifying T cell states that correspond to immunologically relevant surface proteins[12,32]. Szabo et al.[33] used a variant of non-negative matrix factorization (NMF) to identify seven gene expression modules in T cells (e.g. cytotoxicity, proliferation, and IFN response). In a recent study, Jagadeesh et al.[34] identified gene programs through NMF, and found significant heritability enrichment for celiac disease in the putative enhancers of a T cell program characterized by *ETS1, CD247*, and *CD28*. Close collaboration between experimental, statistical, and computational scientists will be essential to interpret gene modules suggested by modern scRNAseq datasets in the context of long-established T cell functions. Using these gene modules to represent T cell state may uncover more eQTLs that are context-dependent. Ongoing work in our group seeks to define the genetic regulation of these functional modules, and to what extent they colocalize with autoimmune disease risk.

## Genetic regulation of the immunological synapse (HLA-TCR)

One of the most critical T cell functions is antigen recognition through the immunological synapse[35]. Fragments of antigen presented on HLA molecules are recognized by the TCR expressed on the surface of T cells. On both sides of the immunological synapse, genetic variation has important consequences. On the antigen presentation side, the MHC locus, which encodes HLA proteins, is among the most polymorphic in the human genome. Germline genetic variation in the MHC locus alters the sequence of HLA proteins, which in turn constrains which antigenic peptides are presented to the immune system. On the T cell side, the gene segments that encode the TCR somatically rearrange separately in each T cell. Newly rearranged TCRs are screened in the thymus, where their affinity to peptides presented on the host's HLA molecules must fall within an optimal range that allows recognition of foreign antigen whilst limiting self-reactivity[36].

How do HLA variants influence thymic selection? Recent work by our group[9] cast TCR amino acid usage as a quantitative molecular trait, and used multivariate regression to search for QTLs in the MHC locus.

Due to extreme linkage disequilibrium, the MHC locus is routinely excluded from QTL studies. However, with careful statistical approaches designed to capture the effects of HLA haplotypes[37], the MHC locus can be robustly investigated for QTLs[38]. In our cdr3-QTL study[9], the strongest associations linked amino acid usage in CDR3, the antigen-recognizing region of the TCR, to HLA-DRB1 amino acid position 13 ("cdr3QTL", Fig. 1b). HLA-DRB1 amino acid position 13 confers a large fraction of genetic risk for both RA, T1D, and other autoimmune diseases. cdr3QTLs raise the intriguing possibility that the thymic selection of autoreactive TCRs plays a critical role in autoimmune disease.

As studies from our group and others identify the effect of disease alleles on TCR amino acids, it becomes important to understand how those TCR amino acid changes alter antigen recognition, the likelihood of T cell activation, and the resultant T cell transcriptional state. Predicting cognate antigen from TCR sequence is a well-recognized problem, currently limited by available training data. Progress in this space will depend critically on whether new TCR de-orphanization technologies[39] can expand the available training data to capture the range of antigens recognized by each TCR. Our recent work[40,41] has demonstrated a role for TCR amino acid features in guiding differentiation of memory and regulatory T cell states. Perhaps, genetic variants in the MHC region that confer risk for autoimmune disease promote the thymic selection of TCRs inclined toward an effector memory T cell state. Consistent with this hypothesis, Orrù et al.[31] reported a csaQTL for the relative proportion of effector memory CD4+ T cells near *HLA-DRB1* (rs9271536).

## Future directions

Building a functional interpretation of a GWAS largely consists of two tasks: linking loci to genes, and identifying the critical cell state(s). Identifying disease-critical cell states based on GWAS is a complex challenge, but recent progress has been made with multimodal scATAC-RNAseq[42]. Efforts to link loci to genes have largely focused on genetic variation between individuals, but limited sample sizes restrict the power of these approaches. As a promising alternative approach, we and others have recently leveraged variation between cells in terms of chromatin accessibility (multimodal scATAC-RNAseq) to construct

**Table 1 | Useful resources for the study and interpretation of immune cell QTLs**

| Name | Description | URL |
|---|---|---|
| ImmPort[51] | Data sharing portal for published data, encompassing transcriptomic, metabolomic, proteomic, and flow cytometry-based profiling of immune cells | https://www.immport.org/home |
| dbSNP[52] | Broad collection of genetic polymorphisms, documenting the rsID number, genomic position, reference and alternate alleles, and possible clinical significance of each polymorphism | https://www.ncbi.nlm.nih.gov/snp/ |
| GWAS Catalog[53] | Database with interactive browser for GWAS summary statistics, aggregating results from over 50,000 GWA studies. | https://www.ebi.ac.uk/gwas/ |
| eQTL Catalogue[54] | eQTL and sQTL summary statistics from uniform processing of 32 published datasets, including bulk RNAseq and scRNAseq. | https://www.ebi.ac.uk/eqtl/ |
| GTEx[55] Portal | Database with interactive browser for the Genotype-Tissue Expression project, which applied bulk RNAseq to 54 non-diseased tissue sites from nearly 1000 individuals. Includes scRNAseq samples from 16 donors for 8 tissues. Open access to eQTL summary statistics and gene count data; access to sequencing data requires application through dbGaP | https://gtexportal.org/home/ |
| DICE[17] Database | Database with interactive browser for the DICE project, which applied bulk RNAseq to 15 FACS-sorted immune cell states (including 11 T cell states) from 91 individuals. Interactive browser allows users to query by gene, rsID, or cell state to collect, visualize and download cell-state-dependent eQTL summary statistics. | https://dice-database.org |
| IMGT[56] | The International Immunogenetics Information System. Documents international nomenclature, nucleotide and amino acid sequences for HLA, TCR and Immunoglobulin genes across 38 species | https://www.imgt.org/ |
| QTLbase2[57] | Curated summary statistics from 377 independent QTL studies and 22 types of molecular QTLs, including eQTLs, pQTLs, caQTLs, and sQTLs. | http://mulinlab.org/qtlbase |
| scQTLbase[58] | Database with interactive browser for eQTLs identified in single-cell datasets. Includes a web tool that runs R package "coloc"[59] to test for colocalization between the existing single-cell QTLs and user-supplied GWAS summary statistics for a trait of interest. | http://bioinfo.szbl.ac.cn/scQTLbase |

gene-enhancer maps[43,44] that can link GWAS variants to causal genes and cell types.

High-throughput screening technologies may drastically expand our understanding of genetic risk variants. Massively parallel reporter assays (MPRA) can simultaneously screen thousands of GWAS variants for regulatory activity, and has recently pinpointed which RA-associated variant in the *BACH2* locus actually reduces expression of *BACH2*, promoting effector T cell differentiation[45]. However, MPRA assays do not account for chromatin inaccessibility, leading to potential false positive results. CRISPR-Cas9 screening approaches, on the other hand, can incorporate chromatin accessibility profiling, and have recently been applied to primary T cells[46]. Future work should extend these genome editing approaches to introduce disease-relevant genetic variants in primary T cells in a high-throughput manner, and characterize the molecular traits that result. This would avoid the complexity of linkage between potential causal variants. Unconstrained by natural selection, these technologies could open the door to molecular characterization of highly pathogenic variants.

Molecular characterizations of disease-associated loci will continue to gain complexity (Table 1). Synthesizing results across different types of QTLs presents a new challenge that will soon become critical. For example, how should we interpret a disease-associated locus that appears to regulate the chromatin accessibility of a gene (caQTL), but not the gene's expression (*cis*-eQTL)? Bossini-Castillo et al. suggest that we have not yet identified the relevant cell state for the *cis*-eQTL[21]. Another possibility, however, is that the locus is a *cis*-eQTL, but its association test just falls short of statistical significance due to limited power. Statistical methods that boost colocalization power by integrating evidence across multiple '-omic' layers are starting to emerge. The Bayesian method OPERA[47], for example, identified 58% more genes relevant to complex traits after considering six types of QTLs (e.g. caQTLs, mQTLs, pQTLs) in conjunction with eQTLs. We are eager to see this approach extended to T cell functional traits, with appropriate modeling of cell state dependence.

Tissue-resident T cells may play a crucial role in the development of autoimmunity. Studies examining T cells from inflamed tissues have identified disease-relevant T cell states that were not previously appreciated: for example, *IL17*+ CD8 T cells in UC[48] and *GZMK*+ T cells in RA[49]. Single-cell profiling of systemic lupus erythematosus (SLE) patient samples recently identified specific regulation of *ORMDL3* at a previously difficult to annotate SLE locus[50]. Hence additional emphasis on collecting tissue samples from patients with autoimmune disease and directly sampling tissue sites of inflammation is critical. Similarly, new studies should query genomic variants that modify the cellular response to disease-relevant stimuli. Identifying multiple types of QTLs (eQTLs, pQTLs, cdr3QTLs) relevant to T cell function in inflamed tissue will bring us closer to understanding how genetic variation shapes risk for autoimmune disease.

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

## Competing interests

The authors declare no competing interests.
