## [Peer Review File · Nature Communications]

REVIEWERS' COMMENTS:

Reviewer #1 (expert in computational and statistical biology, human genomics, immunology and autoimmunity):

This is a Comment from Dr. Roychaudhuri's group that summarizes recent studies on how genetic variants may impact T cell functional traits, a potential mechanism for their relation to autoimmunity. We think this is an important contribution and will provide a great resource for immunologists, human geneticists and computational scientists interested in immunology. We provide our suggestions below, which we believe may further improve the manuscript towards becoming a more authoritative reference for the field.

Specific comments

1. If allowed in this article format, we suggest that authors use another figure or make a new panel in Figure 1 that shows schematic for other types of QTLs relevant to T cells that are discussed here. This will set up the review nicely.
2. When introducing the SNP rs3087243 and CTLA4:
 - a. authors should mention the reference and alternate alleles of rs3087243
 - b. Provide a one-liner for significance of CTLA4
 - c. Authors should mention the exact distance between the SNP and gene,
 - d. Multi-step mechanisms in trans-eQTL: need to provide some potential such mechanisms, maybe 1-2 sentences. The same goes for cis, cis effects are also sufficiently complex that "directly" may not be a good term to refer to them
3. Page 4, they should replace the term "colocalize" and use overlap/coincide since colocalization now refers to a separate statistical analysis to identify the shared variants jointly associated with two traits. Also, the link between the lack of overlapping SNPs and cell-specificity is not clear (page 5, 1st paragraph).
4. Although the authors mentioned that eQTLs and GWAS SNPs rarely coincide (page 4), they can be in LD as well. It is not clear whether the 25% considers LD or not. Also, 25% is a large number to call "rarely".
5. Authors should include a broader set of QTL literature and their functional characterization, including some of the references below:
 - a. Bossini-Castillo et al. Cell Genomics 2022 (PMID: 35591976) – multiple types of QTLs.
 - b. Mouri et al. Nat Genet 2022 (PMID: 35513721) – validated SNPs by MPRA.
 - c. Ohkura et al. Immunity 2020 (PMID: 32362325) – related to Fig. 1a.
 - d. Van Der Wijst et al. Nat Genet 2018 (PMID: 29610479)
 - e. Yazar et al. Science 2022 (PMID: 35389779)
6. Page 5, the authors mention that "accounting for T cell states has substantially enhanced disease-relevant eQTL discovery in T cells". It would be great if they provided more specific information on what fraction of these cell-state-specific eQTLs are actually disease-relevant and are missed by bulk RNA-seq.
7. Page 6, another type of QTL studied in T cells (by BLUEPRINT and others) is histone modification QTLs. These should be referenced for completeness.
8. Page 7, can a few more sentences be said about morphology QTLs to inform the readers about how such QTLs are derived?
9. Page 7, can the authors put their perspective regarding the integration of cell-state (or cell-type-specific) eQTLs (authors should also include the reference Kim-Hellmuth et al. Science 2020; PMID: 32913075) with the gene expression modules (reference 16) or functional modules enriched for heritability (reference 17)?
10. Many eQTL / GWAS studies omit the HLA regions from analysis. Authors need to put their comments regarding using HLA region-specific QTLs (page 8).
11. In the conclusion, it is not clear how multiple types of QTLs (eQTLs, pQTLs, cdrQTLs) would be jointly analyzed for cell-specificity and disease relevance. Do the authors suggest a simple overlap between them or refer to specific multimodal/multivariate analysis? More pointers would be great for the field.

Minor

- each which -> each of which

- between statisticians and immunologists -> between computational scientists and immunologists?
- future studies should consider introducing genetic variation -> mention that this has to be high throughput
- The term "tissue-resident" can be used in the last paragraph

Ferhat Ay and Sourya Bhattacharyya

Reviewer #2 (expert in bioinformatics, functional genomics and genetics of complex diseases):

The manuscript, "The genetic basis of autoimmunity seen through the lens of T cell functional traits" by Kaitlyn A. Lagattuta et al., addresses a critical issue in autoimmune disease research: the genetic underpinnings and the role of T cell functional traits in these diseases. The authors have integrated data from genome-wide association studies (GWASs) and functional trait analysis to provide a comprehensive and insightful exploration of this subject matter. Their investigation of T cell functional traits within the context of autoimmunity and the resulting discussion on cell type-specific eQTLs, cdr3QTLs, and the role of T cell states enrich the existing literature in this field. There are a few minor concerns that could be addressed:

1. The manuscript would benefit from a broader discussion on other types of QTLs, such as caQTLs, hQTLs, meQTL, etc. An article by Lara Bossini-Castillo et al., "Immune disease variants modulate gene expression in regulatory CD4+ T cells" (<https://doi.org/10.1016/j.xgen.2022.100117>), delves into these QTL types and their implications for understanding the regulation of CD4+ T cells. Incorporating insights from this study could enrich your analysis and discussion.
2. The manuscript describes several cell state abundance QTLs (csaQTLs). Including a visual representation, such as a figure illustrating a csaQTL, could enhance readers' comprehension of this type of QTL and its implications.
3. In the "Future directions" section, consider discussing the identification of causal cell types as a crucial part of the functional interpretation of GWAS data. While numerous computational methods have been developed for this purpose, it remains a challenging and important area of research.
4. It would be beneficial for the authors to mention existing important resources, databases, or tools that could aid readers in further understanding the genetic basis of autoimmunity and T cell functional traits.

Reviewer #3 (expert in rheumatology, rheumatoid arthritis disease susceptibility and bioinformatics):

Lagattuta et al present a well written, clear, concise but comprehensive overview of functional traits studies in T cells, with a focus on QTLs, and explain how they can be utilised to interpret GWAS findings. This topic is very relevant for scientists working in autoimmune diseases, given the focus on T cells, but it will also be of interest to the wider community in the area of complex traits genetics.

The authors do a great job at covering the most recent findings relating to the topic and, importantly, at hinting at the complexity involving molecular traits e.g. the importance of T cell states and the need to investigate patients' samples from inflamed tissues.

My suggestion would be to expand the discussion around the apparent lack of co-localization between eQTLs and GWAS variants (this seems very relevant, if the goal is to functionally interpret GWAS variants; is the co-localization better in single cell datasets from patients' tissue?), and perhaps include a comment on recent work by Franke et al regarding co-expression QTLs, which adds to the above-mentioned complexity.

RESPONSE LETTER

We thank the reviewers for providing thoughtful and constructive feedback on the manuscript. In response to their comments, we have made substantial improvements to the manuscripts, which we organize below in a point-by-point response.

Reviewer #1 Comment 1

If allowed in this article format, we suggest that authors use another figure or make a new panel in Figure 1 that shows schematic for other types of QTLs relevant to T cells that are discussed here. This will set up the review nicely.

RESPONSE:

We thank the reviewer for this suggestion. We agree that an additional figure to illustrate other types of QTLs would enhance the comprehensiveness of our manuscript:

new Figure 2. Quantitative Trait Loci (QTL) can regulate diverse molecular phenotypes

Figure 2. Schematic illustrating molecular phenotypes that could be affected by a hypothetical quantitative trait locus (QTL). At this hypothetical locus, an individual may have a TT genotype (in purple), an AA genotype (in orange), or be heterozygous. Along the top row, we see molecular consequences of genotype TT. Along the bottom row, we see molecular consequences of genotype AA. We depict six types of QTLs as examples; this set of six is not comprehensive.

From left to right: For the hypothetical caQTL, individuals with the TT genotype exhibit more closed chromatin than individuals with the AA genotype. For the hypothetical meQTL, individuals with the TT genotype exhibit less DNA methylation of CpG island X compared to individuals with the AA genotype. For the hypothetical eQTL, individuals with the TT genotype exhibit greater expression of gene Y compared to individuals with the AA genotype. For the hypothetical sQTL, RNA splicing in individuals with the TT genotype retains all exons in the transcript of gene Y, while RNA splicing in individuals with the AA genotype excludes the middle exon from the transcript of gene Y. For the hypothetical cQTL, individuals with the TT genotype exhibit higher levels of cytokine Z (in blue) compared to individuals with the AA genotype. For

the hypothetical *csaQTL*, individuals with the TT genotype exhibit a greater relative abundance of the yellow cell state compared to individuals with the AA genotype.

Reviewer #1 Comment 2.1

When introducing the SNP rs3087243 and CTLA4:

- A. authors should mention the reference and alternate alleles of rs3087243
- B. Provide a one-liner for significance of CTLA4
- C. Authors should mention the exact distance between the SNP and gene →

RESPONSE:

We thank the reviewer for these detailed recommendations. We have updated the manuscript text accordingly:

***updated* Main text:**

For example, rs3087243 (reference allele: “G”, alternate allele: “A”): an individual can have 0, 1, or 2 copies of the “G” allele. Each additional copy of the “G” allele corresponds to a decrease in the expression of *CTLA4* in T cells^{4,5}. As an inhibitory receptor, CTLA4 is a vital negative regulator of T cell activation⁶. Since rs3087243 is located within one Mb of the *CTLA4* gene body (1.1 kB from the coding sequence),

Reviewer #1 Comment 2.2

When introducing the SNP rs3087243 and CTLA4:

- D. Multi-step mechanisms in trans-eQTL: need to provide some potential such mechanisms, maybe 1-2 sentences. The same goes for cis, cis effects are also sufficiently complex that “directly” may not be a good term to refer to them.

RESPONSE:

We thank the reviewer for recommending that we add more detail in our definitions of *cis*- and *trans*-eQTL effects. We have updated the manuscript text to include an example mechanism through which a variant could impact expression of a *cis*-gene and an example mechanism through which a variant could impact expression of a *trans*-gene:

***updated* Main text:**

Located near their target genes, *cis*-eQTLs impact expression often by altering transcription factor binding in a proximal regulatory element, or altering the rate of mRNA degradation. In contrast, *trans*-eQTLs, which are located farther away from their target genes, can emerge from a wide range of mechanisms. *trans*-eQTLs may regulate distant gene targets by first altering the expression of a nearby

transcription factor (*cis*-eQTL mediation)⁸. However, *trans*-eQTLs can also emerge from unusual mechanisms. For example, *trans*-eQTLs in the major histocompatibility (MHC) locus on chromosome 6 affect the expression of TCR genes on chromosomes 7 and 14^{9,10}. Most likely, these *trans*-eQTLs change the set of antigenic peptides bound and presented by HLA, which in turn shapes the thymic selection of TCRs.

Reviewer #1 Comment 3.1

Page 4, they should replace the term “colocalize” and use overlap/coincide since colocalization now refers to a separate statistical analysis to identify the shared variants jointly associated with two traits.

RESPONSE:

We agree that it is essential to disambiguate overlapping genomic loci from a formal statistical test for colocalization. For this reason, we use the term “coincide” rather than “colocalize” on page 4:

However, most risk variants for autoimmune disease do not coincide with the eQTLs identified through bulk RNAseq.

For completeness, we have taken another look at all four uses of the term “colocalize” in the manuscript to ensure that they each refer to a formal statistical test. They do indeed all refer to applications of either coloc (Giambartolomei et al. 2013) or JLIM (Chun et al. 2017).

Reviewer #1 Comment 3.2

Also, the link between the lack of overlapping SNPs and cell-specificity is not clear (page 5, 1st paragraph).

RESPONSE:

We thank the reviewer for pointing out this lack of clarity. We have added text to further explain this connection:

***updated* Main text:**

However, bulk RNAseq aggregates gene expression across all cells in a given sample. It is possible that eQTLs relevant to disease only occur in pathogenic subsets of cells, and are obscured in bulk RNAseq aggregation.

Reviewer #1 Comment 4

Although the authors mentioned that eQTLs and GWAS SNPs rarely coincide (page 4), they can be in LD as well. It is not clear whether the 25% considers LD or not. Also, 25% is a large number to call “rarely”.

RESPONSE:

We thank the reviewer for pointing out this lack of clarity. We have added text to further explain this connection:

updated Main text:

However, most risk variants for autoimmune disease do not coincide with the eQTLs identified through bulk RNAseq. With a statistical approach that accounts for linkage disequilibrium (LD), only 25% of autoimmune GWAS associations appear to share underlying genetic causes (“statistical colocalization”) with eQTL results from bulk RNAseq¹⁴.

Reviewer #1 Comment 5

Authors should include a broader set of QTL literature and their functional characterization, including some of the references below:

- a. Bossini-Castillo et al. Cell Genomics 2022 (PMID: 35591976) – multiple types of QTLs.
- b. Mouri et al. Nat Genet 2022 (PMID: 35513721) – validated SNPs by MPRA.
- c. Ohkura et al. Immunity 2020 (PMID: 32362325) – related to Fig. 1a.
- d. Van Der Wijst et al. Nat Genet 2018 (PMID: 29610479)
- e. Yazar et al. Science 2022 (PMID: 35389779)

RESPONSE:

We thank the reviewer for providing a broader list of QTL studies. We have incorporated each of these references into manuscript:

updated Main text:

T_{reg} cells, while highly relevant to immune-mediated disease, constitute a rare cell state (less than 5% of peripheral T cells¹⁸). Thus, it will be important to continue focused isolation and analysis of the T_{reg} cell population for eQTL discovery¹⁹ (**Bossini-Castillo et al. 2022**).

updated Main text:

For example, how should we interpret a disease-associated locus that appears to regulate the chromatin accessibility of a gene (caQTL) but not the gene’s expression (*cis*-eQTL)? **Bossini-Castillo et al.** suggest that we have not yet identified the relevant cell state for the *cis*-eQTL¹⁹.

updated Main text:

High-throughput screening technologies may drastically expand our understanding of genetic risk variants. Massively parallel reporter assays (MPRA) can simultaneously screen thousands of GWAS variants for regulatory activity, and have

recently pinpointed which RA-associated variant in the *BACH2* locus actually reduces expression of *BACH2*, promoting effector T cell differentiation⁴³ (**Mouri et al. 2022**).

updated Main text:

rs7731626 corresponded to a larger increase in *IL6ST* expression in regulatory T cells (T_{regs}) compared to non-regulatory T cells (**Figure 1a**). Evidence that the functional consequences of rs7731626 may be concentrated in T_{regs} could extend to many other autoimmune-associated SNPs, which are known to be enriched in certain naive T_{reg} -specific regulatory regions of the genome^{1,2,17} (**Ohkura et al. 2020**).

updated Main text:

There are multiple ways to represent how eQTL effects depend on transcriptional context. We use dimensionality reduction to identify major transcriptional gradients that each approximate a cell state such as T_{reg} . We then identify cell-state-dependent eQTLs, where a genotype's effect on a gene's expression is modulated by the transcriptional gradient⁵. Because transcriptional gradients may be tagged by key marker genes, however, it is sometimes possible to reframe cell-state-dependent eQTLs as co-expression QTLs (co-eQTLs)^{20,21} (**Van der Wijst et al. 2018, Li et al. 2023**). In the co-eQTL framing, the correlation in expression between gene A and gene B depends on a genotype. An alternative way to describe this phenomenon is that the genotype's effect on gene A depends on the expression of gene B. If gene B is expressed in a specific cell state, cell-state-dependent eQTL and co-eQTL are synonymous. If gene B does not tag a cell state, the locus is not a cell-state-dependent eQTL, but would still be considered a co-eQTL. The co-eQTL framework, therefore, offers a more inclusive interpretation of gene expression interaction.

updated Main text:

In a similar study, **Yazar***, **Alquicira-Hernandez***, **Wing* et al.**²³ applied scRNAseq to PBMCs at steady state, and scanned for eQTLs that colocalized with genetic risk for autoimmune disease. They observed that 68% of the T cell eQTLs colocalizing with disease loci were detected in only one of the T cell states.

Reviewer #1 Comment 6

Page 5, the authors mention that “accounting for T cell states has substantially enhanced disease-relevant eQTL discovery in T cells”. It would be great if they provided more specific information on what fraction of these cell-state-specific eQTLs are actually disease-relevant and are missed by bulk RNA-seq.

RESPONSE:

We thank the reviewer for this suggestion. We agree this is one of the central takeaways of the piece, and are happy to provide more details.

updated Main text:

Multiple studies have recently estimated that a substantial portion of eQTLs depend on cell state and that this proportion increases when considering autoimmune disease loci. Soskic*, Cano-Gamez* et al.²² profiled CD4+ T cells over a time course of anti-CD3/anti-CD28 stimulation, and found that 2,265 of 6,407 (35%) eQTLs depended on the activation state of the T cell. They then queried whether each eQTL statistically colocalized with genetic risk for immune-mediated disease. 60% of the colocalizing eQTLs were specific to T cell activation, and would have been missed if they did not account for this cell state dependence. In a similar study, Yazar*, Alquicira-Hernandez*, Wing* et al.²³ applied scRNAseq to PBMCs at steady state, and scanned for eQTLs that colocalized with genetic risk for autoimmune disease. They observed that 68% of the T cell eQTLs colocalizing with disease loci were detected in only one of the T cell states. Evidently, accounting for T cell states has substantially enhanced disease-relevant eQTL discovery in T cells. The majority of autoimmune GWAS associations, however, still remain unexplained.

Reviewer #1 Comment 7

Page 6, another type of QTL studied in T cells (by BLUEPRINT and others) is histone modification QTLs. These should be referenced for completeness.

RESPONSE:

We thank the reviewer for this suggestion.

updated Main text:

Other types of QTLs are steadily gaining recognition, such as chromatin accessibility QTLs (caQTLs), histone modification QTLs (hQTLs), DNA methylation QTLs (meQTLs), splicing QTLs (sQTLs), and protein QTLs (pQTLs) (Figure 2).

Reviewer #1 Comment 8

Page 7, can a few more sentences be said about morphology QTLs to inform the readers about how such QTLs are derived?

RESPONSE:

We thank the reviewer for this suggestion. This type of QTL is indeed quite different from the others, and warrants a bit more explanation:

updated Main text:

We and others have identified cell morphology QTLs (cmQTLs)²⁵ by measuring thousands of morphological phenotypes (e.g. size of mitochondria, granularity of endoplasmic reticulum) in vitro through multiplexed staining and imaging. cmQTLs

can connect genetic variants to cellular function in human-derived, disease-relevant cell types.

Reviewer #1 Comment 9

Page 7, can the authors put their perspective regarding the integration of cell-state (or cell-type-specific) eQTLs (authors should also include the reference Kim-Hellmuth et al. Science 2020; PMID: 32913075) with the gene expression modules (reference 16) or functional modules enriched for heritability (reference 17)?

RESPONSE:

We thank the reviewer for these suggestions. We have incorporated a reference to Kim-Hellmuth et al. Science 2020 to motivate cell-type-specific eQTLs:

***updated* Main text:**

It is possible that eQTLs relevant to disease only occur in pathogenic subsets of cells, and are obscured in bulk RNAseq aggregation. Consistent with this hypothesis, dividing bulk RNAseq tissue samples into constituent cell types by flow cytometry¹⁵ or *in silico* deconvolution¹⁶ has nominated more eQTLs which are significantly enriched for disease-associated loci.

We have also added a few comments regarding our perspective on the integration of cell-state-specific eQTLs with gene expression modules:

***updated* Main text:**

Close collaboration between experimental, statistical, and computational scientists will be essential to interpret gene modules suggested by modern scRNAseq datasets in the context of long-established T cell functions. **Using these gene modules to represent T cell state may uncover more eQTLs that are context dependent.** Ongoing work in our group seeks to define the genetic regulation of these functional modules, **and to what extent they colocalize with autoimmune disease risk.**

Reviewer #1 Comment 10

Many eQTL / GWAS studies omit the HLA regions from analysis. Authors need to put their comments regarding using HLA region-specific QTLs (page 8).

RESPONSE:

We thank the reviewer for this suggestion. Our laboratory is indeed quite interested in QTL discovery for the MHC region, and we are happy to include pointers in this direction:

updated Main text:

Due to extreme linkage disequilibrium, the MHC locus is routinely excluded from QTL studies. However, with careful statistical approaches designed to capture the effects of HLA haplotypes³⁵, the MHC locus can be robustly investigated for QTLs³⁶.

Reviewer #1 Comment 11

In the conclusion, it is not clear how multiple types of QTLs (eQTLs, pQTLs, cdrQTLs) would be jointly analyzed for cell-specificity and disease relevance. Do the authors suggest a simple overlap between them or refer to specific multimodal/multivariate analysis? More pointers would be great for the field.

RESPONSE:

We thank the reviewer for this suggestion. We agree that it is not obvious how to proceed, and have spent some time thinking about the approaches that are emerging. We have added the following paragraph to the Future Directions section:

updated Main text:

Molecular characterizations of disease-associated loci will continue to gain complexity. Synthesizing results across different types of QTLs presents a new challenge that will soon become critical. For example, how should we interpret a disease-associated locus that appears to regulate the chromatin accessibility of a gene (caQTL) but not the gene's expression (*cis*-eQTL)? Bossini-Castillo et al. suggest that we have not yet identified the relevant cell state for the *cis*-eQTL¹⁹. Another possibility, however, is that the locus *is* a *cis*-eQTL; its association test just falls short of statistical significance due to limited power. Statistical methods that boost colocalization power by integrating evidence across multiple -omic layers are starting to emerge. For example, the Bayesian method OPERA⁴⁵ identified 58% more genes relevant to complex traits after considering six types of QTLs (e.g. caQTLs, mQTLs, pQTLs) in conjunction with eQTLs. We are eager to see this approach extended to T cell functional traits, with appropriate modeling of cell state dependence.

Reviewer #1 Comment 12

each which -> each of which

RESPONSE:

We thank the reviewer for catching this grammatical error. We have corrected the text accordingly:

updated Main text:

In a recent demonstration of the power of cQTLs, Nath et al²⁸ analyzed 11 circulating cytokines, each of which may functionally represent the coordinated activity of thousands of genes.

Reviewer #1 Comment 13

between statisticians and immunologists -> between computational scientists and immunologists?

RESPONSE:

We thank the reviewer for this suggestion. We have expanded the terminology as following:

***updated* Main text:**

Close collaboration between **experimental, statistical, and computational scientists** will be essential to interpret gene modules suggested by modern scRNAseq datasets in the context of long-established T cell functions

Reviewer #1 Comment 13

future studies should consider introducing genetic variation -> mention that this has to be high throughput

RESPONSE:

We thank the reviewer for this suggestion. We modified the text accordingly:

***updated* Main text:**

Future work should extend these genome editing approaches to introduce **disease-relevant** genetic variants in primary T cells **in a high throughput manner**, and **characterize** the molecular traits that result.

Reviewer #1 Comment 14

The term "tissue-resident" can be used in the last paragraph

RESPONSE:

We thank the reviewer for this suggestion. We have modified the text accordingly:

***updated* Main text:**

Tissue-resident T cells may play a crucial role in the development of autoimmunity.

Reviewer #2 Comment 1

The manuscript would benefit from a broader discussion on other types of QTLs, such as caQTLs, hQTLs, meQTL, etc. An article by Lara Bossini-Castillo et al., "Immune disease variants modulate gene expression in regulatory CD4+ T cells"

(<https://doi.org/10.1016/j.xgen.2022.100117>), delves into these QTL types and their implications for understanding the regulation of CD4+ T cells. Incorporating insights from this study could enrich your analysis and discussion.

RESPONSE:

We thank the reviewer for this reference and suggestion. We have expanded our coverage of QTL types, while keeping the piece focused on T cells and autoimmunity:

updated Main text:

Other types of QTLs are steadily gaining recognition, such as chromatin accessibility QTLs (caQTLs), **histone modification QTLs (hQTLs)**, **DNA methylation QTLs (meQTLs)**, splicing QTLs (sQTLs), and protein QTLs (pQTLs) (**Figure 2**). These QTLs are **conceptually analogous to eQTLs, with gene expression substituted for some other molecularly-defined trait; particular approaches are comprehensively reviewed elsewhere²⁴.**

We reviewed Bossini-Castillo et al. with great interest, and incorporated this reference into the manuscript in multiple places:

updated Main text:

T_{reg} cells, while highly relevant to immune-mediated disease, constitute a rare cell state (less than 5% of peripheral T cells¹⁸). Thus, it will be important to continue focused isolation and analysis of the T_{reg} cell population for eQTL discovery¹⁹ (Bossini-Castillo et al. 2022).

updated Main text:

For example, how should we interpret a disease-associated locus that appears to regulate the chromatin accessibility of a gene (caQTL) but not the gene's expression (*cis*-eQTL)? **Bossini-Castillo et al. suggest that we have not yet identified the relevant cell state for the *cis*-eQTL¹⁹.**

Reviewer #2 Comment 2

The manuscript describes several cell state abundance QTLs (csaQTLs). Including a visual representation, such as a figure illustrating a csaQTL, could enhance readers' comprehension of this type of QTL and its implications.

RESPONSE:

We thank the reviewer for this suggestion. We have designed an additional figure that includes a visual depiction of csaQTLs:

new Figure 2. Quantitative Trait Loci (QTL) can regulate diverse molecular phenotypes

Figure 2. Schematic illustrating molecular phenotypes that could be affected by a hypothetical quantitative trait locus (QTL). At this hypothetical locus, an individual may have a TT genotype (in purple), an AA genotype (in orange), or be heterozygous. Along the top row, we see molecular consequences of genotype TT. Along the bottom row, we see molecular consequences of genotype AA. We depict six types of QTLs as examples; this set of six is not comprehensive.

From left to right: For the hypothetical caQTL, individuals with the TT genotype exhibit more closed chromatin than individuals with the AA genotype. For the hypothetical meQTL, individuals with the TT genotype exhibit less DNA methylation of CpG island X compared to individuals with the AA genotype. For the hypothetical eQTL, individuals with the TT genotype exhibit greater expression of gene Y compared to individuals with the AA genotype. For the hypothetical sQTL, RNA splicing in individuals with the TT genotype retains all exons in the transcript of gene Y, while RNA splicing in individuals with the AA genotype excludes the middle exon from the transcript of gene Y. For the hypothetical cQTL, individuals with the TT genotype exhibit higher levels of cytokine Z (in blue) compared to individuals with the AA genotype. For the hypothetical csaQTL, individuals with the TT genotype exhibit a greater relative abundance of the yellow cell state compared to individuals with the AA genotype.

Reviewer #2 Comment 3

In the "Future directions" section, consider discussing the identification of causal cell types as a crucial part of the functional interpretation of GWAS data. While numerous computational methods have been developed for this purpose, it remains a challenging and important area of research.

RESPONSE:

We thank the reviewer for this observation. As suggested, we have added some context to the Future Directions section:

updated Main text:

Building a functional interpretation of a GWAS largely consists of two tasks: linking loci to genes, and identifying critical cell state(s). Identifying disease-critical cell states based on GWAS is a complex challenge, but recent progress has been made with multimodal scATAC-RNAseq⁴⁰.

Reviewer #2 Comment 4

It would be beneficial for the authors to mention existing important resources, databases, or tools that could aid readers in further understanding the genetic basis of autoimmunity and T cell functional traits

RESPONSE:

We thank the reviewer for their insightful suggestion. There is certainly an abundance of tools and databases relevant to this topic, so we have compiled a short list of those we consider most useful. We hope this additional table will provide a more practical guide for those interested in beginning research in this area.

new **Table 1.** Useful resources for the study and interpretation of immune cell QTLs

Name	Description	URL
ImmPort ⁴⁹	Data sharing portal for published data, encompassing transcriptomic, metabolomic, proteomic, and flow cytometry-based profiling of immune cells.	https://www.immport.org/home
dbSNP ⁵⁰	Broad collection of genetic polymorphisms, documenting the rsID number, genomic position, reference and alternate alleles, and possible clinical significance of each polymorphism.	https://www.ncbi.nlm.nih.gov/snp/
GWAS Catalog ⁵¹	Database with interactive browser for GWAS summary statistics, aggregating results from over 50 thousand GWA studies.	https://www.ebi.ac.uk/gwas/
eQTL Catalogue ⁵²	eQTL and sQTL summary statistics from uniform processing of 32	https://www.ebi.ac.uk/eqt/

	published datasets, including bulk RNAseq and scRNAseq.	
GTEX ⁵³ Portal	Database with interactive browser for the Genotype-Tissue Expression project, which applied bulk RNAseq to 54 non-diseased tissue sites from nearly 1000 individuals. Includes scRNAseq samples from 16 donors for 8 tissues. Open access to eQTL summary statistics and gene count data; access to sequencing data requires application through dbGaP.	https://gtexportal.org/home/
DICE ¹⁵ Database	Database with interactive browser for the DICE project, which applied bulkRNAseq to 15 FACS-sorted immune cell states (including 11 T cell states) from 91 individuals. Interactive browser allows users to query by gene, rsID, or cell state to collect, visualize and download cell-state-dependent eQTL summary statistics.	https://dice-database.org
IMGT ⁵⁴	The International Immunogenetics Information System. Documents international nomenclature, nucleotide and amino acid sequences for HLA, TCR and Immunoglobulin genes across 38 species.	https://www.imgt.org/
QTLbase2 ⁵⁵	Curated summary statistics from 377 independent QTL studies and 22 types of molecular QTLs, including eQTLs, pQTLs, caQTLs, and sQTLs.	http://mulinlab.org/qtlbase

scQTLbase ⁵⁶	Database with interactive browser for eQTLs identified in single cell datasets. Includes a colocalization web tool that runs R package “coloc” ⁵⁷ to test for colocalization between the existing single cell QTLs and user-supplied GWAS summary statistics for a trait of interest.	http://bioinfo.szbl.ac.cn/scQTLbase
--	---

Reviewer #3 Comment 1

My suggestion would be to expand the discussion around the apparent lack of co-localization between eQTLs and GWAS variants (this seems very relevant, if the goal is to functionally interpret GWAS variants; is the co-localization better in single cell datasets from patients’ tissue?)

RESPONSE:

We thank the reviewer for this suggestion. We agree this is one of the main discussion points of the piece, and have expanded our remarks in two places. First, with respect to improved colocalization in single cell analyses accounting for cell state:

updated Main text:

Multiple studies have recently estimated that a substantial portion of eQTLs depend on cell state and that this proportion increases when considering autoimmune disease loci. Soskic*, Cano-Gamez* et al.²² profiled CD4+ T cells over a time course of anti-CD3/anti-CD28 stimulation, and found that 2,265 of 6,407 (35%) eQTLs depended on the activation state of the T cell. They then queried whether each eQTL statistically colocalized with genetic risk for immune-mediated disease. 60% of the colocalizing eQTLs were specific to T cell activation, and would have been missed if they did not account for this cell state dependence. In a similar study, Yazar*, Alquicira-Hernandez*, Wing* et al.²³ applied scRNAseq to PBMCs at steady state, and scanned for eQTLs that colocalized with genetic risk for autoimmune disease. They observed that 68% of the T cells eQTLs colocalizing with disease were detected in only one of the T cell states. Evidently, accounting for T cell states has substantially enhanced disease-relevant eQTL discovery in T cells. The majority of autoimmune GWAS associations, however, still remain unexplained.

In response to the review’s point regarding patient tissue, we’ve added to our Future Directions section:

updated Main text:

Studies examining T cells from inflamed tissues have identified disease-relevant T cell states that were not previously appreciated: for example, *IL17+* CD8 T cells in UC⁴⁶ and *GZMK+* T cells in RA⁵⁸. Single-cell profiling of Systemic Lupus Erythematosus (SLE) patient samples recently identified specific regulation of *ORMDL3* at a previously difficult to annotate SLE locus. Hence additional emphasis on collecting samples from patients with autoimmune disease and directly sampling tissue sites of inflammation is critical.

Reviewer #3 Comment 2

and perhaps include a comment on recent work by Franke et al regarding co-expression QTLs, which adds to the above-mentioned complexity.

RESPONSE:

We thank the reviewer for this suggestion. We too have been intrigued by Franke et al.'s formulation of gene expression interactions, and how they relate to our approach. We find cell state-dependent eQTLs and co-expression QTLs to be remarkably synonymous, and have added a section to the manuscript describing their relationship. We believe this will provide valuable clarity, and we are grateful for the recommendation!

***updated* Main text:**

There are multiple ways to represent how eQTL effects depend on transcriptional context. We use dimensionality reduction to identify major transcriptional gradients that each approximate a cell state such as T_{reg} . We then identify cell-state-dependent eQTLs, where a genotype's effect on a gene's expression is modulated by the transcriptional gradient⁵. Because transcriptional gradients may be tagged by key marker genes, however, it is sometimes possible to reframe cell-state-dependent eQTLs as co-expression QTLs (co-eQTLs)^{20,21}. In the co-eQTL framing, the correlation in expression between gene A and gene B depends on a genotype. An alternative way to describe this phenomenon is that the genotype's effect on gene A depends on the expression of gene B. If gene B is expressed in a specific cell state, cell-state-dependent eQTL and co-eQTL are synonymous terms. If gene B does not tag a cell state, the locus is not a cell-state-dependent eQTL, but would still be considered a co-eQTL. The co-eQTL framework, therefore, offers a more inclusive interpretation of gene expression interaction. The vast number of possible pairings between genes and their regulatory loci precludes comprehensive detection of co-eQTLs, which would be required to estimate the proportion of eQTLs that are co-eQTLs. Alternatively, by focusing on cell states rather than individual genes, we are able to estimate that approximately one third (33%) of eQTLs in T cells depend on transcriptional context⁵.

References

1. Trynka, G. *et al.* Chromatin marks identify critical cell types for fine mapping complex trait variants. *Nat. Genet.* **45**, 124–130 (2013).
2. Amariuta, T. *et al.* IMPACT: Genomic Annotation of Cell-State-Specific Regulatory Elements Inferred from the Epigenome of Bound Transcription Factors. *Am. J. Hum. Genet.* **104**, 879–895 (2019).
3. Hu, X. *et al.* Integrating autoimmune risk loci with gene-expression data identifies specific pathogenic immune cell subsets. *Am. J. Hum. Genet.* **89**, 496–506 (2011).
4. Chen, L. *et al.* Genetic Drivers of Epigenetic and Transcriptional Variation in Human Immune Cells. *Cell* **167**, 1398–1414.e24 (2016).
5. Nathan, A. *et al.* Single-cell eQTL models reveal dynamic T cell state dependence of disease loci. *Nature* **606**, 120–128 (2022).
6. Walunas, T. L. *et al.* CTLA-4 can function as a negative regulator of T cell activation. *Immunity* **1**, 405–413 (1994).
7. Pai, A. A. *et al.* The contribution of RNA decay quantitative trait loci to inter-individual variation in steady-state gene expression levels. *PLoS Genet.* **8**, e1003000 (2012).
8. Westra, H.-J. *et al.* Systematic identification of trans eQTLs as putative drivers of known disease associations. *Nat. Genet.* **45**, 1238–1243 (2013).
9. Sharon, E. *et al.* Genetic variation in MHC proteins is associated with T cell receptor expression biases. *Nat. Genet.* **48**, 995–1002 (2016).
10. Ishigaki, K. *et al.* HLA autoimmune risk alleles restrict the hypervariable region of T cell receptors. *Nat. Genet.* **54**, 393–402 (2022).
11. Ueda, H. *et al.* Association of the T-cell regulatory gene CTLA4 with susceptibility to autoimmune disease. *Nature* **423**, 506–511 (2003).
12. Okada, Y. *et al.* Genetics of rheumatoid arthritis contributes to biology and drug discovery.

- Nature* **506**, 376–381 (2014).
13. Onengut-Gumuscu, S. *et al.* Fine mapping of type 1 diabetes susceptibility loci and evidence for colocalization of causal variants with lymphoid gene enhancers. *Nat. Genet.* **47**, 381–386 (2015).
 14. Chun, S. *et al.* Limited statistical evidence for shared genetic effects of eQTLs and autoimmune-disease-associated loci in three major immune-cell types. *Nat. Genet.* **49**, 600–605 (2017).
 15. Schmiedel, B. J. *et al.* Impact of Genetic Polymorphisms on Human Immune Cell Gene Expression. *Cell* **175**, 1701–1715.e16 (2018).
 16. Kim-Hellmuth, S. *et al.* Cell type-specific genetic regulation of gene expression across human tissues. *Science* **369**, (2020).
 17. Ohkura, N. *et al.* Regulatory T Cell-Specific Epigenomic Region Variants Are a Key Determinant of Susceptibility to Common Autoimmune Diseases. *Immunity* **52**, 1119–1132.e4 (2020).
 18. Baecher-Allan, C., Viglietta, V. & Hafler, D. A. Human CD4+CD25+ regulatory T cells. *Semin. Immunol.* **16**, 89–98 (2004).
 19. Bossini-Castillo, L. *et al.* Immune disease variants modulate gene expression in regulatory CD4+ T cells. *Cell Genom* **2**, None (2022).
 20. van der Wijst, M. G. P. *et al.* Single-cell RNA sequencing identifies celltype-specific cis-eQTLs and co-expression QTLs. *Nat. Genet.* **50**, 493–497 (2018).
 21. Li, S. *et al.* Identification of genetic variants that impact gene co-expression relationships using large-scale single-cell data. *Genome Biol.* **24**, 80 (2023).
 22. Soskic, B. *et al.* Immune disease risk variants regulate gene expression dynamics during CD4+ T cell activation. *Nat. Genet.* **54**, 817–826 (2022).
 23. Yazar, S. *et al.* Single-cell eQTL mapping identifies cell type-specific genetic control of autoimmune disease. *Science* **376**, eabf3041 (2022).

24. Zhu, Z. *et al.* Leveraging molecular quantitative trait loci to comprehend complex diseases/traits from the omics perspective. *Hum. Genet.* **142**, 1543–1560 (2023).
25. Tegtmeyer, M. *et al.* High-dimensional phenotyping to define the genetic basis of cellular morphology. *bioRxiv* 2023.01.09.522731 (2023) doi:10.1101/2023.01.09.522731.
26. Huang, Q. Q., Ritchie, S. C., Brozynska, M. & Inouye, M. Power, false discovery rate and Winner's Curse in eQTL studies. *Nucleic Acids Res.* **46**, e133–e133 (2018).
27. Zlotnik, A. Perspective: Insights on the Nomenclature of Cytokines and Chemokines. *Front. Immunol.* **11**, 908 (2020).
28. Nath, A. P. *et al.* Multivariate Genome-wide Association Analysis of a Cytokine Network Reveals Variants with Widespread Immune, Haematological, and Cardiometabolic Pleiotropy. *Am. J. Hum. Genet.* **105**, 1076–1090 (2019).
29. Orrù, V. *et al.* Complex genetic signatures in immune cells underlie autoimmunity and inform therapy. *Nat. Genet.* **52**, 1036–1045 (2020).
30. Nathan, A. *et al.* Multimodally profiling memory T cells from a tuberculosis cohort identifies cell state associations with demographics, environment and disease. *Nat. Immunol.* **22**, 781–793 (2021).
31. Szabo, P. A. *et al.* Single-cell transcriptomics of human T cells reveals tissue and activation signatures in health and disease. *Nat. Commun.* **10**, 4706 (2019).
32. Jagadeesh, K. A. *et al.* Identifying disease-critical cell types and cellular processes by integrating single-cell RNA-sequencing and human genetics. *Nat. Genet.* **54**, 1479–1492 (2022).
33. Grakoui, A. *et al.* The immunological synapse: a molecular machine controlling T cell activation. *Science* **285**, 221–227 (1999).
34. Ashby, K. M. & Hogquist, K. A. A guide to thymic selection of T cells. *Nat. Rev. Immunol.* (2023) doi:10.1038/s41577-023-00911-8.
35. Sakaue, S. *et al.* Tutorial: a statistical genetics guide to identifying HLA alleles driving

- complex disease. *Nat. Protoc.* (2023) doi:10.1038/s41596-023-00853-4.
36. Kang, J. B. *et al.* Mapping the dynamic genetic regulatory architecture of HLA genes at single-cell resolution. *medRxiv* (2023) doi:10.1101/2023.03.14.23287257.
 37. Dobson, C. S. *et al.* Antigen identification and high-throughput interaction mapping by reprogramming viral entry. *Nat. Methods* **19**, 449–460 (2022).
 38. Lagattuta, K. A. *et al.* Repertoire analyses reveal T cell antigen receptor sequence features that influence T cell fate. *Nat. Immunol.* **23**, 446–457 (2022).
 39. Lagattuta, K. A., Nathan, A., Rumker, L., Birnbaum, M. E. & Raychaudhuri, S. The T cell receptor sequence influences the likelihood of T cell memory formation. *bioRxiv* (2023) doi:10.1101/2023.07.20.549939.
 40. Gupta, A. *et al.* Dynamic regulatory elements in single-cell multimodal data capture autoimmune disease heritability and implicate key immune cell states. *bioRxiv* (2023) doi:10.1101/2023.02.24.23286364.
 41. Dey, K. K. *et al.* SNP-to-gene linking strategies reveal contributions of enhancer-related and candidate master-regulator genes to autoimmune disease. *Cell Genom* **2**, (2022).
 42. Sakaue, S. *et al.* Tissue-specific enhancer-gene maps from multimodal single-cell data identify causal disease alleles. *bioRxiv* (2022) doi:10.1101/2022.10.27.22281574.
 43. Mouri, K. *et al.* Prioritization of autoimmune disease-associated genetic variants that perturb regulatory element activity in T cells. *Nat. Genet.* **54**, 603–612 (2022).
 44. Legut, M. *et al.* A genome-scale screen for synthetic drivers of T cell proliferation. *Nature* **603**, 728–735 (2022).
 45. Wu, Y. *et al.* Joint analysis of GWAS and multi-omics QTL summary statistics reveals a large fraction of GWAS signals shared with molecular phenotypes. *Cell Genom* **3**, 100344 (2023).
 46. Smillie, C. S. *et al.* Intra- and inter-cellular rewiring of the human colon during ulcerative colitis. *Cell* **178**, 714–730.e22 (2019).

47. Jonsson, A. H. *et al.* Granzyme K+ CD8 T cells form a core population in inflamed human tissue. *Sci. Transl. Med.* **14**, eabo0686 (2022).
48. Perez, R. K. *et al.* Single-cell RNA-seq reveals cell type-specific molecular and genetic associations to lupus. *Science* **376**, eabf1970 (2022).
49. Bhattacharya, S. *et al.* ImmPort: disseminating data to the public for the future of immunology. *Immunol. Res.* **58**, 234–239 (2014).
50. Sherry, S. T. *et al.* dbSNP: the NCBI database of genetic variation. *Nucleic Acids Res.* **29**, 308–311 (2001).
51. MacArthur, J. *et al.* The new NHGRI-EBI Catalog of published genome-wide association studies (GWAS Catalog). *Nucleic Acids Res.* **45**, D896–D901 (2017).
52. Kerimov, N. *et al.* A compendium of uniformly processed human gene expression and splicing quantitative trait loci. *Nat. Genet.* **53**, 1290–1299 (2021).
53. GTEx Consortium. The GTEx Consortium atlas of genetic regulatory effects across human tissues. *Science* **369**, 1318–1330 (2020).
54. Lefranc, M.-P. IMGT, the International ImMunoGeneTics Information System. *Cold Spring Harb. Protoc.* **2011**, 595–603 (2011).
55. Huang, D. *et al.* QTLbase2: an enhanced catalog of human quantitative trait loci on extensive molecular phenotypes. *Nucleic Acids Res.* **51**, D1122–D1128 (2023).
56. Ding, R. *et al.* scQTLbase: an integrated human single-cell eQTL database. *Nucleic Acids Res.* (2023) doi:10.1093/nar/gkad781.
57. Giambartolomei, C. *et al.* Bayesian test for colocalisation between pairs of genetic association studies using summary statistics. *PLoS Genet.* **10**, e1004383 (2014).
58. Zhang, F. *et al.* Defining inflammatory cell states in rheumatoid arthritis joint synovial tissues by integrating single-cell transcriptomics and mass cytometry. *Nat. Immunol.* **20**, 928–942 (2019).

REVIEWERS' COMMENTS:

Reviewer #1 (expert in computational and statistical biology, human genomics, immunology and autoimmunity):

The authors have done a fantastic job in addressing all our comments.

Reviewer #2 (expert in bioinformatics, functional genomics and genetics of complex diseases):

The authors have addressed my previous comments adequately.

Reviewer #3 (expert in rheumatology, rheumatoid arthritis disease susceptibility and bioinformatics):

The authors have satisfactorily addressed all my comments and I have no further concerns.